# FoMo: A unifying theory of visual foraging

**Alasdair D. F. Clarke** [ID]*, **Anna E. Hughes** [ID]

Department of Psychology, University of Essex, Colchester, United Kingdom

* a.clarke@essex.ac.uk

## Abstract

Visual foraging lies at the intersection of visual perception, decision-making and action planning. An attractive feature of this paradigm is that it generates a rich stream of sequential decision data. However, this presents a number of challenges for analysis. To this end, we have developed FoMo, a robust and flexible generative model for spatial-sequential data that allows prediction of participants' selection behaviour on a target-by-target basis. Building upon our initial work, we present an updated version of FoMo (Clarke, et al. PLOS Comput Biol. 18:e1009813, 2022), which incorporates spatial structure allowing us to model organised spatial behaviours. FoMo provides estimates of a range of interpretable parameters, meaning we can use it to understand the causes of behavioural differences: for example, incorporating spatial-structure parameters improves model prediction accuracy for a number of visual foraging datasets, predominantly due to improvements for a subset of participants who use grid-following strategies. Our approach can also account for individual differences across the wide range of descriptive statistics that have previously been used to explore human and non-human animal behaviour, providing a unified framework for analysing these data.

## Author summary

Foraging is an important behaviour for humans and other animals, helping them to find food, habitats, and mates. It can involve searching both through visual space (such as searching for berries on a bush) and abstract space (such as when retrieving a word from memory). Here, we update our computational model of visual foraging behaviour (FoMo) which allows us to predict the behaviour of foragers in tasks where they must sequentially collect multiple targets. In particular, we allow the model to predict organised spatial behaviours, such as using a 'reading-like' strategy where participants search systematically in horizontal directions. Importantly, this means that we can study individual differences in foraging strategies in a quantitative, principled manner. We also show that our model can act as a unifying framework for visual foraging research, as it is able

**Data availability statement:** All data used in the manuscript are available at https://github.com/Riadsala/FoMo/tree/main/data (and this repository is linked to in the manuscript). In addition, all datasets used are secondary and are openly available in other repositories on the internet as well. Kristjánsson et al: https://doi.org/10.1371/journal.pone.0100752.s001. Tagu et al: https://doi.org/10.17605/OSF.IO/E7CND. Clarke et al: https://osf.io/y6qbv/ (data can be found on the 'Github' tab). Hughes et al: https://github.com/Riadsala/foraging_scarcity.

**Funding:** The author(s) received no specific funding for this work.

**Competing interests:** The authors have declared that no competing interests exist.

to predict a wide range of different descriptive statistics that have been used in previous work.

## 1 Introduction

When a person searches, they search through space. This is true whether we are talking about a classic cognitive psychology paradigm in which an observer searches for a pre-specified target on a computer screen [1] or more 'real-world' tasks, such as finding a specific wine bottle in a virtual store [2]. It even holds for abstract semantic spaces that are searched through when receiving a word from memory [3,4]. The effect of space is particularly relevant in foraging behaviours; a topic of study in human and non-human animal cognition [5], as well as more widely in ecology and biology [6]. We define foraging as a task in which participants sequentially collect multiple targets that are distributed throughout some (physical or abstract) space. The distance from the forager to the prospective items is generally a strong predictor of behaviour [7]: people will generally select a closer target in preference to one further away, all other things being equal.

While the above may seem obvious, the effect of space on search tasks is frequently treated as a nuisance parameter to be "averaged away". Instead of trying to incorporate spatial parameters into modelling efforts, it is more common to attempt to reduce their influence via experimental design. For example, when designing stimuli for visual search, a common approach is to arrange items in a circle around a central fixation cross in order to control for any effects of eccentricity. The effects of direction are averaged away by ensuring that there are multiple trials with the target in randomised positions in each one. Some illustrative examples include [8–12], but we could cite many hundreds of other papers, including many of our own. Similarly, some visual foraging paradigms use stimulus displays with constantly moving items [13] in order to disrupt, remove or at least minimise the spatial factors. The hope is that these manipulations allow researchers to experimentally isolate the particular factors that they are investigating, and for many research questions this approach has been extremely fruitful. However, we would argue that in building a full model of visual foraging, we need to grapple with spatial effects more directly. Incorporating spatial position explicitly in models allows us to not only partition out this variance, but also to understand how spatial position might interact with other factors, such as stimulus features. Finally, because spatial factors strongly influence how people behave on this task, we need to include them to be able to generate useful predictions of behaviour on an individual level.

This introduction is structured as follows. First, we discuss the effect of space in the visual attention literature, with a focus on how visual search models account for spatial effects. We then move to the specific case of visual foraging and introduce the key paradigms. Finally, we discuss the various descriptive statistics and modelling approaches that have been used to analyse data from visual foraging experiments. We purposely have kept this literature review relatively brief: For more in-depth

recent reviews, see [5] and [14]. This general introduction is followed by Section 2 in which we introduce our generative model of visual foraging (FoMo, [7]) and extend it to create a unifying model that incorporates target-based as well as spatial parameters.

## 1.1 Eccentricity, eye movements & attention

The importance of space in vision is well established. For a start, the density of rods and cones on the retina (and hence visual acuity) falls off steeply with increasing eccentricity from the fovea [15]. The effect of this is further compounded by M-scaling, in which the amount of visual cortex devoted to processing each unit of visual space decreases systematically with eccentricity [16,17]. The result is a dramatic gradient of spatial resolution across the visual field, with the highest acuity concentrated in a small central region and steep drop-offs toward the periphery.

One way in which the human visual system deals with these eccentricity effects is to make use of saccadic eye movements [18], and a number of spatial effects have been well documented over the years. For example, phenomena such as saccadic momentum (the tendency of saccades to continue in the same direction and with the same velocity as the previous saccade, [19,20]) and inhibition of return (the reduced probability of a saccade returning to a previously fixated location [21,22]) play a role in making some regions of the stimulus more likely to be fixated than others. It is also possible in many cases to detect regularities in the direction of saccades. Gilchrist and colleagues [23] demonstrated that scan paths during visual search contain more horizontal than vertical saccades, and that this ratio is sensitive to spatial regularities in the search array: participants made more horizontal saccades in trials where the stimuli had a stronger grid structure. Finally, there are behavioural biases such as the central bias [24], where observers favour making fixations towards the centre of an image, regardless of content, and pseudoneglect, the tendency to be biased towards the left-hand side of space [25]. Despite these factors being well known, there are few attempts to account for them or even measure them in visual search tasks, although the saccadic flow model is one attempt to provide a relatively simple model of how people look around photographs of scenes which takes into account the central bias [26,27].

In addition to eye movements, our visual system employs spatial attention to selectively prioritise different regions of the visual field, enhancing detection sensitivity and discrimination accuracy in attended areas [28]. However, despite the name emphasising the *spatial* nature of attention, much of the empirical work aims to investigate theoretical accounts of how attention may be deployed away from the fovea, without much explicit reference to the distance or direction. As mentioned above, the variables are often controlled for, in the hope that there the effects under investigation do not interact with direction or distance. We discuss this in more depth in the context of visual search below.

## 1.2 Spatial effects in visual search

The most influential models of visual search are generally relatively unconcerned with spatial features. The only mention in Feature Integration Theory (FIT) [1] is to argue that there is no requirement to spatially locate target features. More recent iterations of the Guided Search (GS) Model [29] highlight the importance of spatial information and acknowledge "*in classic GS, spatial factors were largely ignored.*" However, the discussion is largely limited to the functional field of view (i.e., the area around the current fixation that defines the current spatial limits of search), with little focus on the effects of spatial layout on the order in which items are inspected. Finally, ideal observer models of visual search [30] address the optimal distributions of saccadic directions and distances, but do not account for individual spatial search behaviour [31–33].

One well-understood spatial effect is the fall off in visibility with increasing eccentricity: it is almost trivial to vision scientists that all other things being equal, a target that is further from fixation will take longer to find than one that is closer, even in the case where an observer is carrying out a parallel, 'pop out' search that is thought to occur simultaneously across the visual field [11]. However, the importance of spatial effects goes beyond this. For example, search slopes vary with eccentricity, particularly when the target is distinguished from the distractors by shape [12]. There are also other

studies that demonstrate broader spatial effects in visual search tasks. For example, participants often use a 'reading-like' strategy when completing a visual search task [34], and systematic scanning can happen around circular displays, suggesting that these biases do not simply reflect oculomotor biases but instead are related to the display structure [35]. These types of scanning strategies can also have practical consequences: for example, when searching through 3D medical images organised into layers, radiologists who adopted a 'scanning' strategy, searching an entire 2D scan before moving in depth had a lower true positive rate for detecting nodules compared to participants who had a 'drilling' strategy, restricting their eye movements to one portion of the image and scrolling through in depth [36]. However, these types of strategies are rarely considered in models of search, despite potentially explaining a high proportion of the variance in people's searching behaviours.

## 1.3 Visual foraging

It is clear that spatial factors strongly affect visual search, and we therefore need a paradigm that allows us to examine the spatial components of search behaviour easily. We believe that the visual foraging paradigm, first developed by Kristjánsson and colleagues [37], is particularly suitable for this. In this task, participants are directed to sequentially collect targets (usually of multiple classes, e.g., red circles and green circles) by clicking on them, while ignoring distractors (e.g., blue circles and yellow circles). Thus, we are able to obtain from each trial a sequence of target selections that give us information about how participants spatially explore the environment. This paradigm has the advantage of being constrained and relatively tractable, as in the basic paradigm there are fewer than 100 simple items on the screen. In addition, a range of previous work has demonstrated that participants' strategies can vary. For example, Kristjánsson and colleagues [37] showed that people's behaviour in a feature foraging task (where targets are defined from distractors by just one feature) differs compared to a conjunction foraging task (where the targets must be distinguished from distractors using a combination of factors). In particular, when there are two classes of targets, most people switch more frequently between target types in the feature condition compared to the conjunction condition, generating longer 'run' lengths in the latter (although there are individual differences in this behaviour, and it appears to be less prominent in 3D environments [38,39]). There is also good evidence that participants can plan ahead during foraging, preparing the next one or two targets that they will visit [40,41]. In our own work [42], we have also seen that participants clearly use a range of strategies, leading to interesting individual differences in target selection patterns (see Fig 1).

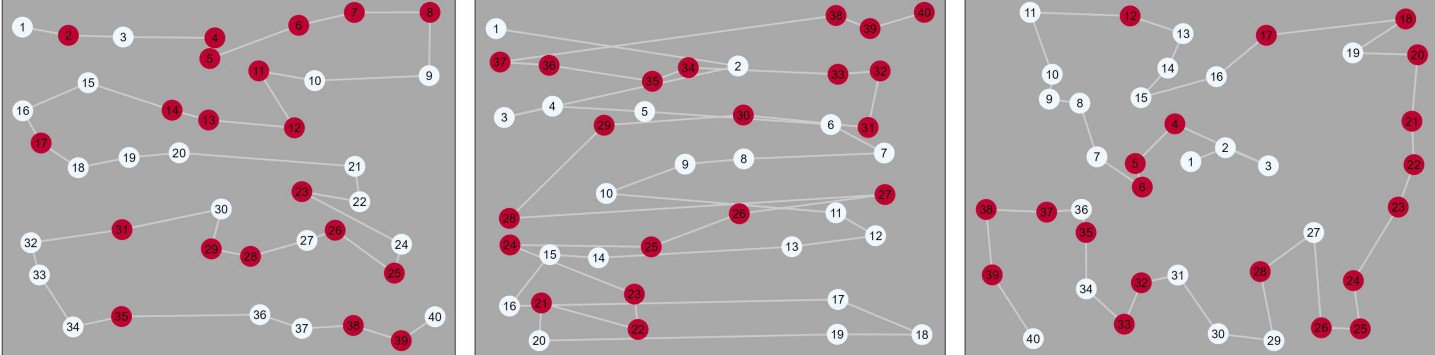

**Fig 1. Examples of three trials from the [42] dataset.** Each trial is from the same condition (*feature*) but with a different participant. We can clearly see that there are some large individual differences. The first two participants appear to prioritise a horizontal scanning strategy, although the second implements this as two long runs (one colour, and then the next). The third participant de-emphasies a directional strategy and instead focuses on collecting nearby items of the same colour whenever possible.

Spatial factors within the context of foraging have also been explored. In one VR study, human participants organised their search behaviour based on the layout of the area to be searched [43]. Similarly, human participants often used highly systematic search paths when attempting to find small targets (coins) in an open terrain search area [44] Interestingly, other primate species (such as capuchin monkeys) are also able to forage 'patchy' stimuli in an efficient manner [45], but only humans were able to efficiently search a less naturalistic matrix stimulus [43], indicating that some level of spatially organised foraging may be a highly evolutionarily conserved ability, perhaps driven by diet [46] (as primates eat a diet rich in fruit, which tends to be ephemeral and patchily distributed). However, humans seem to be able to flexibly adapt these strategies to serve a wider range of scenarios, and understanding which strategies they choose in different circumstances is critical for understanding foraging behaviour more fully.

## 1.4 Models and measurements of foraging

Foraging paradigms are a rich source of data and can be analysed in many different ways. In the animal cognition and ecology literature, foraging is often studied in terms of *patch leaving*: when should the local area be considered depleted, triggering a move to a new, hopefully, more bountiful, region? This has often been studied via the lens of the *marginal value theorem* (MVT) [47], which states that an animal should leave a patch when the marginal capture rate dips below the average rate for its habitat. This is a specific example of the more general *optimal foraging theory* [48]. These frameworks have been used to study foraging behaviour in a wide range of species [5], albeit with limited success in humans: a recent review by [14] concluded that these MVT methods and patch-leaving statistics provided a poor account of human visual behaviour.

Another concept from the ecology foraging literature is the idea of *lévy flights* which have been used to describe movement patterns in animals as varied as the albatross [49] and polar bears [50] (see [51] for a general review). In short, lévy flight analyses aim to describe foraging behaviour as a random walk, and assume that the distribution of inter-target distances are sampled from a power law. As with MVT, these ideas have also been taken up by researchers studying foraging in humans [52]. However, such analysis has limitations: as lévy flight analysis only attempts to summarise the distribution on inter-target distances, it is not able to offer predictions of which items are selected next. Furthermore, these methods do not incorporate directionality [53], a point we explore in the present manuscript.

Whereas foraging studies from ecology typically focus on explaining patch-leaving or how animals move around their environment, studies on human visual foraging have often focused on explaining behaviour *within* a trial. As the paradigm has gained in popularity, a wealth of descriptive statistics have been applied. As previously discussed, many of these experiments use two different target classes and focus on 'runs' [37,54]: a condition can then be summarised by calculating the average number of runs, or the average run length. This work has inspired a model by Le and colleagues [55] that aims to predict when a participant will break with the current run and switch to the other target type. Their classifier makes use of 11 features (such as the distance to the closest same target, and the ratio of remaining targets of one type to the other type) and aims to predict switches during both feature and conjunction foraging data. Their approach achieves impressive accuracy rates (92.1% for conjunction foraging) and can also predict rare switching events relatively well. However, this type of model cannot yet deal with the individual differences seen in this type of foraging task.

Beyond run statistics, a variety of other measures have been introduced with the aim of capturing something about spatial organisation during foraging [56]. The best-*r* statistic is based on correlating the *x* and *y* coordinates of the selected targets with the order in which they were selected. Trials in which targets were systematically collected from left-to-right or top-to-bottom receive high scores, while more stochastic, less organised behaviour receives lower scores. While this measure appears relatively well established in the literature [57–59] it has several weaknesses that we believe undermine its usefulness as a measure of spatial organisation. First of all, it can only capture horizontal and vertical patterns and is blind to diagonal zig-zag and circular spiral patterns. Secondly, this statistic summarises the whole foraging path, and it is hard to see how it could be used as a causal predictor to help us understand item-to-item selections within a trial. Similarly, we

also have the *percentage above optimal* (*PAO*) and *intersection rate* (*IR*) statistics, both of which are calculated from a completed foraging path. *PAO* compares the total distance travelled by the participant to the shortest possible path, while *IR* counts the number of times the path intersects itself. Again, while both statistics have been used to describe foraging organisation with some success [58,60], they describe only a summary of the whole foraging path and cannot easily be used to make predictions of participant behaviour on an item-by-item basis.

### 1.5 Individual differences in foraging

Visual foraging requires a range of cognitive abilities from low-level attention to higher-order decision making. As such, it is an interesting paradigm to study individual differences. Early work identified potential *super foragers* - a small subset of participants who appeared to be able to carry out the same strategy in both feature and conjunction foraging [37]. However, later work [61] demonstrated that participants who implement this strategy appear to make a larger number of errors, suggesting that these differences may reflect different strategies (perhaps different choices regarding a speed-accuracy trade-off) rather than differences in performance on this task. This conclusion is supported by [42], where low correlations were found between an individual's performance on a visual foraging task when compared with other similar visual attention tasks, suggesting that individuals might display fairly consistent behaviours within a given task, but that these may reflect a chosen strategy rather than an innate cognitive skill that varies between people.

Despite this, there is clear evidence for foraging behaviours varying across the lifespan [56,58,62], with younger participants displaying less organised foraging behaviour (although these differences do not seem to lead to changes in MVT-related measures [63,64]). Similarly, recent work has demonstrated that differences in foraging behaviour may be useful in a clinical context: individuals with higher trait anxiety are more affected by negative emotional stimuli in an 'emotion foraging' task [65], and learning to adjust one's exploration-exploitation balance based on environmental volatility in a patch-leaving foraging task is affected by anxiety [66]. Cancellation tasks are frequently used to assess visuospatial attention deficits in neurological populations [67,68], and involve participants searching for multiple targets among distractors in a manner very similar to a foraging task. It therefore seems likely that visual foraging paradigms are sensitive to developmental differences, and that being able to model behaviour in this task could have benefits for clinical populations by potentially allowing earlier or more sensitive diagnoses.

## 2 Methods

In our work, we have taken a different approach to understanding foraging within a patch (functionally equivalent to a single trial in the datasets used in this manuscript) by conceptualising it as a '*sampling without replacement*' problem [7]. Our Foraging Model (FoMo) provides accurate predictions of target-by-target behaviour using relatively few parameters. Moreover, because our model is generative, we can simulate data to systematically evaluate how well it captures different aspects of human behaviour. This approach allows us to identify specific limitations in the model and use these insights to refine our theoretical understanding of visual foraging.

One weakness of the original version of FoMo [7] is that it made relatively poor predictions of absolute foraging directions (see Fig 2), potentially leading to decreased accuracy in cases where participants use an absolute direction strategy (e.g., follow cardinal directions) to structure their behaviour. In the current manuscript, we update FoMo to account for absolute direction biases, and hence characterise spatial behaviour and participant strategies more accurately. We use a flexible mixture model of directions and show that this improves overall model accuracy, predominantly by dramatically improving accuracy for a subset of individuals who use highly directional strategies to complete foraging tasks. We also demonstrate the flexibility of our model by showing how models with different parameter combinations can easily be compared in our framework, increasing the utility of our approach for a range of different foraging questions. Finally, we show how our modelling approach is able to act as a unifying framework, as it is able to accurately represent the individual differences present in a wide range of other foraging metrics that have been used in previous studies.

PLOS Computational Biology

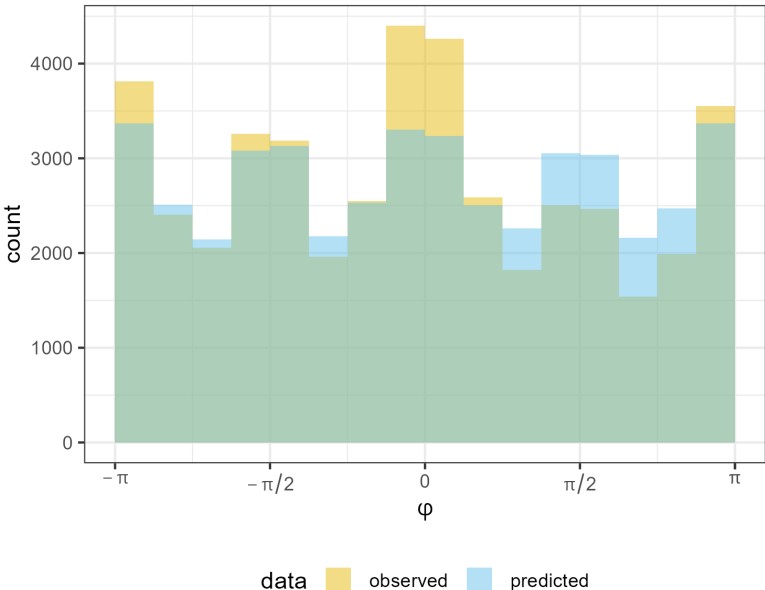

**Fig 2. FoMo v1.0 inter-item selection results for the [42] dataset.** Histograms comparing the absolute directions ($\phi$). Green indicates overlap of overlap of the two distributions. We can see that while the model favours the cardinal directions - presumably due to a proximity bias combined with grid-like stimuli - it fails to capture the strong left to right peak we see in the human data.

On the whole, we find that our model offers a compelling description of behaviour and individual differences in the visual foraging paradigm. The rest of this section contains descriptions of our existing model (both general, Section 2.1, and mathematical, Section 2.2), followed by the new directional components in Section 2.3. Section 2.4 details how the model is implemented, while Section 2.5 gives a brief overview of the datasets we have used to test the model. Finally, Section 2.6 outlines the analysis plan, including the range of foraging metrics we test.

## 2.1 General overview

[7] modelled visual foraging as a weighted sampling-without-replacement process. The foraging model assumes that target items belong to one of two classes (*A* or *B*) and accounts for four influences on how participants decide which item to select next:

- *proximity tuning* ($\rho_\delta$): the extent to which participants select items that are in close proximity to the previously selected item.

- *relative direction tuning* ($\rho_\psi$): the extent to which participants select items that "ahead" of them. Negative values would indicate that items that are "behind" the participant are favoured.

- *stick/switch rate* ($b_s$): values above 0 lead to increasingly long runs of the same target class while values below zero lead to a preference to alternate between target classes.

- *class weights* ($b_a$): This can be thought of as modelling the relative attractiveness of item class *A* over *B*. This attractiveness could be due to a number of different reasons such as low-level visual saliency or value. When $b_a$ is above 0 the model favours selecting items of class *A* while values below 0 lead to a preference for items of class *B*.

These four features are illustrated in Fig 3. These are more formally defined in Section 2.1, and expanded to include absolute directions in Section 2.2.

## 2.2 FoMo v1.0: Model Specification

To formally define FoMo we first define some terms (an overview is provided in Table 1). Let the $N$ target items in a stimulus be represented by $z_i = (x_i, y_i, c_i)$, where $i = 1, \ldots, N$; $(x_i, y_i)$ is the spatial location of the target $z_i$ and $c_i \in \{A, B\}$ represents its class.

### 2.2.1 Item class components.
First, we define some simple functions that give us the weight contributions due to item class (see equation 1) and stick/switching behaviour (see equation 2):

$$p_a(i) = p_a(z_i) = \begin{cases} \text{invlogit}(b_a) & \text{if } c_i = A \\ 1 - \text{invlogit}(b_a) & \text{if } c_i = B \end{cases}$$

(1)

$$p_s(i|j) = p_s(z_i|z_j) = \begin{cases} \text{invlogit}(b_s) & \text{if } c_i = c_j \\ 1 - \text{invlogit}(b_s) & \text{if } c_i \neq c_j \end{cases}$$

(2)

### 2.2.2 Spatial components.
The spatial components of the model are based on the distance, $\delta$, and direction, $\phi$, of the vector from the previously selected item $z_i$ to each remaining item $z_j$:

$$\delta(i, j) = \delta(z_i, z_j) = \sqrt{(x_i - x_j)^2 + (y_i - y_j)^2}$$

(3)

$$\phi(i, j) = \phi(z_i, z_j) = \tan^{-1}(y_j - y_i, x_j - x_i)$$

(4)

We use *relative direction*, $\psi$, defined as the angular difference between the vector from $z_{i-1}$ to $z_i$ and the vectors from $z_i$ to each remaining $z_j$:

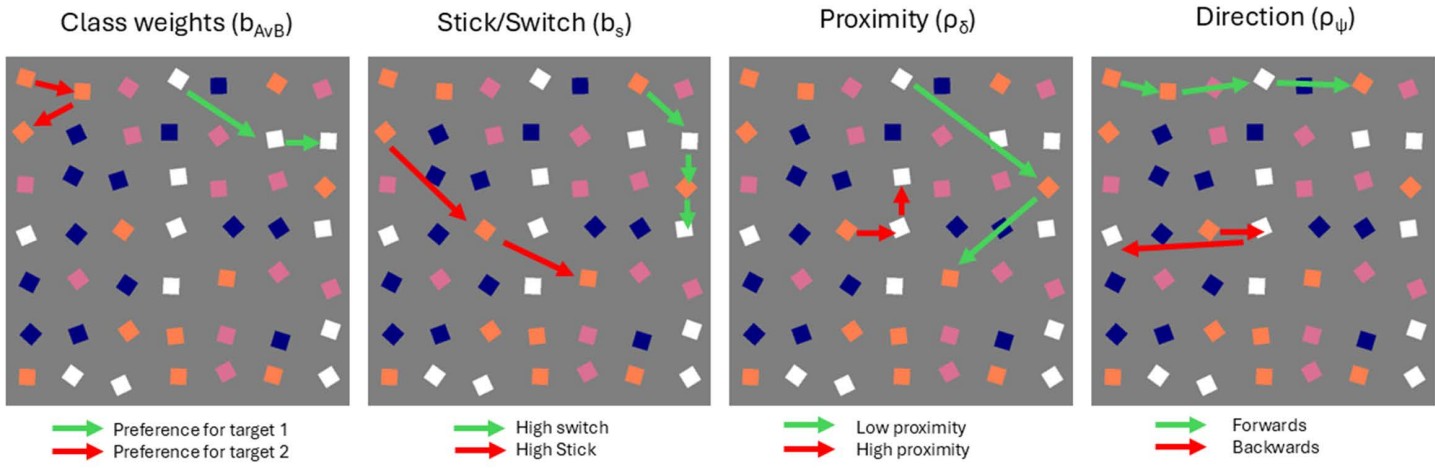

**Fig 3. The original [7] model parameters.** From left to right: stick/switch rate, class weights, proximity tuning, and direction tuning.

**Table 1. Overview of notation used to define FoMo. Concepts are split into four components: stimulus definition; derived features; free parameters; and likelihood calculation.**

| Symbol | Definition | Eq number |
|---|---|---|
| $N$ | number of items in a stimulus | – |
| $z_i, z_j$ | the $i$th and $j$th items in a stimulus | – |
| $z_{i-1}$ | the item selected before item $z_i$ | – |
| $x_i, y_i$ | location of item $z_i$ | – |
| $c_i$ | class of item $z_i$ | – |
| $\delta(i, j)$ | distance from item $z_i$ to $z_j$ | 3 |
| $\delta_0$ | normalising constant | 17 |
| $\phi(i, j)$ | angle from item $z_i$ to $z_j$ | 4 |
| $\psi(i, j)$ | relative direction from $z_i$ to $z_j$ (relative to $z_{i-1}$) | 6 |
| $b_a$ | class weights | 1 |
| $b_s$ | stick/switch rate | 2 |
| $\rho_\delta$ | proximity tuning | 7 |
| $\rho_\psi$ | relative direction tuning | 7 |
| $\kappa_k$ | the concentration of each direction | 10 |
| $\theta_k$ | the relative weighting of the four directions | 10 |
| $p(j|i)$ | the conditional prob. of selecting item $j$ given $i$ | 9 |
| $\omega_{j,i}$ | the weight of selecting item $j$ (unnormalised prob.) | 8, 11 |
| $f_{\theta,\kappa}$ | the sum of four vonMises functions | 10 |
| $k = 1, \ldots, 4$ | an index for the four vonMises functions | 10 |
| $\mu_k$ | the mean of each vonMises functions | 10 |

$$\phi_d(i, j) = \phi(j, i) - \phi(i - 1, i) \tag{5}$$

before a final transform to calculate the shortest angular difference:

$$\psi(i, j) = \min(\left|\phi_d(i, j) \% 2\pi\right|, \left|-\phi_d(i, j) \% 2\pi\right|) \tag{6}$$

This results in a value which equals 1 if the vectors $z_{i-1} \to z_i$ and $z_i \to z_j$ are parallel in the same direction. A value of $\psi = 0.5$ indicates that the vectors are perpendicular, while $\psi = 0$ corresponds to the situation in which $z_{i-1} \to z_i$ and $z_i \to z_j$ are parallel but in opposite directions.

Following [7], we calculate the contribution of $\delta$ and $\psi$ to the item-weights with a negative exponential (see Equation 7). This function was chosen as it allows for the (likely typical) case where there is a strong preference for, e.g., a very nearby item, and then the preference falls off steeply, e.g., items at medium and far distances should be treated similarly by the model.

$$e^{-\rho_\delta \delta(j) - \rho_\psi \psi(j)} \tag{7}$$

**2.2.3 Likelihood.** We can now use these functions to calculate the likelihood of selecting different items. Let $z_i$ ($i = 1, 2, \ldots$) represent the sequence of items that we have selected so far in this trial, while $z_j$ represents the remaining items that have not yet been selected. To simplify the mathematical notation we can define $\delta_j = \delta(z_j, z_i)$ and $\psi_j = \psi(z_{i-1}, z_i, z_j)$. Then weight for selecting each item $z_j$ given our previous selections $z_1, z_2, \ldots, z_i$ is defined by:

$$\omega_{j,i} \sim p_a(j)p_s(j|i)e^{-\rho_\delta \delta_j - \rho_\psi \psi_j} \tag{8}$$

These can be converted into probabilities:

$$p(j|i) = \begin{cases} \frac{\omega(j,i)}{\sum_{z=1}^{N} \omega(z,i)} & \text{if } j \notin \{1, 2, \dots, i\} \\ 0 & \text{if } j \in \{1, 2, \dots, i\} \end{cases} \tag{9}$$

In brief, these formulae tell us that the probability of selecting a particular item depends on its class, the class of the previously selected item, the distance between items, and the difference in directions.

### 2.3 Towards FoMo v2.0: Absolute direction

One limitation of the FoMo v1.0 is that it only cares about relative direction, and is blind to the left-to-right scanning behaviour illustrated in Fig 1. To incorporate these behaviours we have to make use of *absolute direction*, $\phi$. This is somewhat more complex than our relative direction parameter as we now have to make decisions in our modelling about *which* directions we think may be more heavily weighted by human participants, and how best to model these weights.

**2.3.1 Directional components.** Our implementation of absolute directions for FoMo is based on a von Mises mixture model (the von Mises distribution is simply an extension of the normal distribution to work on the circle), illustrated in Fig 4. In short, we assume that items that can be selected by following one of the four cardinal directions ($0, \frac{\pi}{2}, \pi, \frac{3\pi}{2}$) receive greater weight, with items receiving less weight the further they are from these directions. This is done by defining a new function, *f* that computes a score for a given angle $\phi$:

$$f_{\theta,\kappa}(\phi_j) = 1 + \sum_{k=1}^{4} \theta_\psi \texttt{vonMises}(\mu_k, \kappa_k) \tag{10}$$

I.e., for each direction $k$ we need to fit a $\theta_k$, that gives the strength of preference for direction $\mu_k$ and a $\kappa_k$, that gives the concentration (i.e., the spread) around the direction. A high value means that only directions very close to $\mu_k$ receive a large weight, while low values spread the weight over a larger range of directions. We set the $\mu_k$ values to be equal to the four cardinal directions.

We can now adopt our weights calculation to take this new feature into account:

$$\omega_{j,i} \sim p_a(i)p_s(j|i)e^{-\rho_\delta \delta_j - \rho_\psi \psi_j - f_{\theta,\kappa}(\phi_j)} \tag{11}$$

**2.3.2 Constraining $\kappa$.** The inclusion of this directional information adds a further two parameters ($\theta_k, \kappa_k$) for each direction modelled: for example, for model v1.3, with four directional components and two conditions, we add 16 new parameters. Furthermore, jointly fitting $\theta$ and $\kappa$ to the data can be challenging: as $\theta$ approaches 0 (i.e., there is no directional preference), $\kappa$ becomes relatively unconstrained and vice versa (i.e., if $\kappa$ becomes very small, the variance around the mean is very large and $\theta$ is relatively unconstrained). Therefore, we manually set the $\kappa$ values as a hyper-parameter (see Fig 5).

**2.3.3 Model versions.** We will test a number of different versions of FoMo, allowing us to determine whether including absolute direction parameters improves the model, and whether absolute and relative direction parameters both need to be included in the model. The versions we will test are detailed in Table 2.

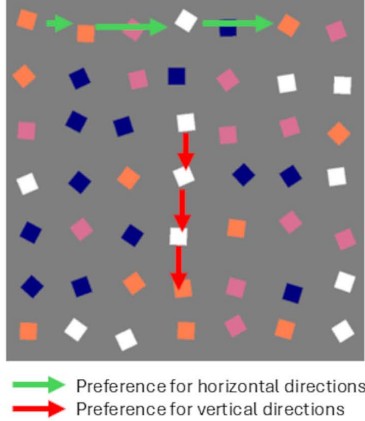

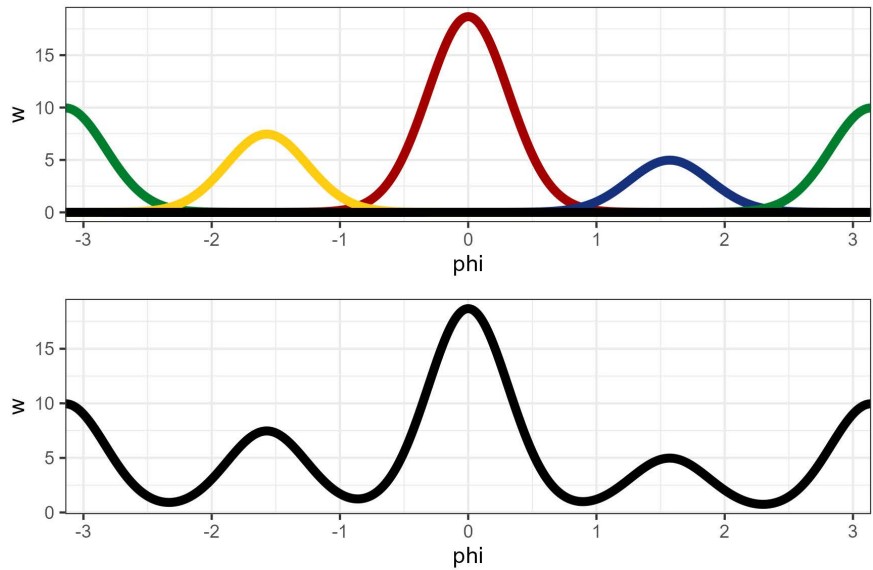

**Fig 4. (top): Illustratifiguon of four von Mises distributions (top) that can be summed to give (middle) a multi-model distribution that is able to flexibly weight horizontal and vertical directions.** (*bottom*): How these weights could encode different strategies.

In order to rigorously test our models we split all tested datasets into separate training and testing partitions. As we are interested in how well the random effect structure of our models can capture differences between participants, we fit the model using the first *N*/2 trials-per-participant-per-condition, and test to the last *N*/2 trials.

## 2.4 Implementation details

This model has been implemented in Stan (a probabilistic programming language) which allows us to estimate the posterior probability for each parameter given observed data. Specifically, we used cmdStan v2.36.0 [69], R v4.4.1 [70] and cmdStanr [71].

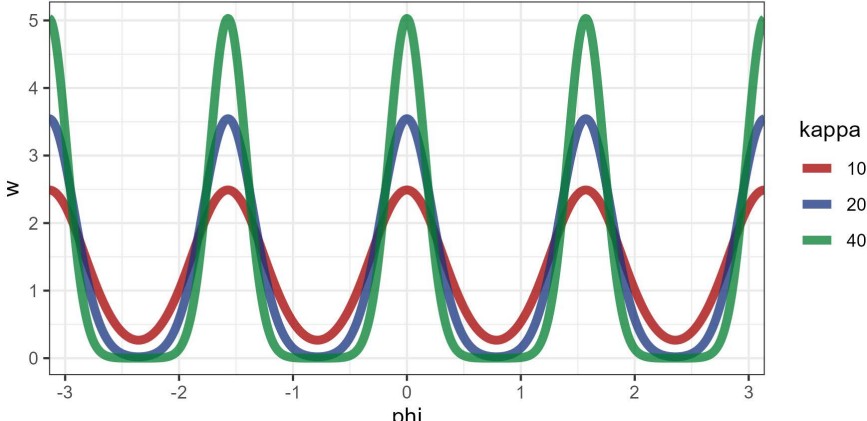

**Fig 5. Illustration of three different $\kappa$ values.** We used a value of $\kappa = 20$ in the work presented here, which strikes a balance between minimising the overlap between our four components without assigning angles around the oblique directions all to 0.

**Table 2. Table with a full overview of the different model versions, detailing which model components are included in each version.**

| version | overview | item class | proximity ($\delta$) | rel dir ($\psi$) | abs dir ($\phi$) |
|---------|----------|-----------|---------------------|------------------|------------------|
| v1.0 | New and improved implementation of model presented in [7] (see Github repository for details of small changes made from previous version) | $b_a$ and $b_s$ | abs | x | |
| v1.1 | Testing an alternative way of computing distance (see Github repository for details) | $b_a$ and $b_s$ | rel | x | |
| v1.2 | Testing whether the relative direction component is important | $b_a$ and $b_s$ | abs | | |
| v1.3 | Testing the addition of four absolute direction components | $b_a$ and $b_s$ | abs | x | 4 |
| v1.4 | Testing the addition of eight absolute direction components | $b_a$ and $b_s$ | abs | x | 8 |
| v1.5 | Testing whether the relative direction component is important for v1.3 | $b_a$ and $b_s$ | abs | | 4 |

**2.4.1 Multi-level framework.** To account for individual differences between participants, we implement a maximal multi-level framework in which each parameter is allowed to vary across conditions and from one participant to the next:

$$b_a \sim \text{condition} + (\text{condition}|\text{person}) \tag{12}$$

$$b_s \sim \text{condition} + (\text{condition}|\text{person}) \tag{13}$$

$$\rho_\delta \sim \text{condition} + (\text{condition}|\text{person}) \tag{14}$$

$$\rho_\psi \sim \text{condition} + (\text{condition}|\text{person}) \tag{15}$$

A full variance-covariance matrix is estimated for these core parameters. As the experiments analysed in this manuscript all have two conditions, this leads to an $8 \times 8$ matrix.

For the directional components we also allow for $\theta$ to have a maximal random effects structure:

$$\theta_k \sim \text{condition} + (\text{condition|person}) \tag{16}$$

However, modelling the full covariance matrix is computationally challenging, especially when we move from four to eight directions, as this would produce a $16 \times 16$ matrix (Initalising highly constrained parameters, such as high dimensional variance-covariance matrices is computationally challenging). As such, we do not model the correlations between distance weights.

**2.4.2 Rescaling distances.** In our previous work we found that the $\rho_\delta$ parameter was typically an order of magnitude larger than the other parameters. This is simply an artifact of the scale that we have used to measure the distances between items in the display. As it is often computationally easier to fit models when the parameters are on similar scales, we have rescaled $\delta$ by dividing all distances by a hyper-parameter $\delta_0$:

$$\delta(z_i, z_j) = \frac{\sqrt{(x_i - x_j)^2 + (y_i - y_j)^2}}{\delta_0} \tag{17}$$

In the models presented in this paper we have used $\delta_0 = 20$.

**2.4.3 Priors.** Based on the results of [7] we use the following weakly-informative priors for the fixed-effect components:

$$b_a \sim \mathcal{N}(0, 1.5) \tag{18}$$

$$b_s \sim \mathcal{N}(0, 1.5) \tag{19}$$

$$\rho_\delta \sim \mathcal{N}(1, 0.25) \tag{20}$$

$$\rho_\psi \sim \mathcal{N}(0, 1) \tag{21}$$

For the random-effects, we follow [72] and use an LKJ prior. We use an Exponential(5) distribution for the prior for all group-level variances. See Github repository for full model code.

**2.4.4 Generated quantities.** To facilitate model comparison by evaluating model predictions we have implemented a *generated quantities* block to our Stan model that computes a range of posterior predictions:

- `log_lik`: the log-likelihood that each item $z_i$ was selected next, given the participant and previous foraging selections in the current trial. While this could in theory be computed for all remaining items in the stimulus, our implementation only computes it for the item that was selected next by the participant.

- *P*: the model's guess of which item will be selected next. This item is randomly sampled from all remaining items using the FoMo weights.

- *Q*: simulating a whole trial from start to finish.

- *F*: the same as *Q*, but with the intial item selection fixed to match the item that was selected in the empirical data.

These values allow us to implement a range of model summary and comparison measures, going beyond the simple posterior density plots used in [7].

## 2.5 Data

Below we give an overview of the datasets we will make use of in this paper: a brief summary is included in Table 3. We used only previously published data sets, so the approval of a research ethics committee was not needed for this manuscript.

**Table 3. Table overviewing the data used in this paper. The 'trials' column indicates the number of trials in each condition, and the 'target items' column indicates how many targets the participant had to collect (all were exhaustive search tasks, apart from [73]). The 'conditions' column indicates the comparisons that were looked at in our analysis, and do not necessarily reflect the full range of conditions that were tested in the original papers.**

| citation | participants | target items | trials | conditions |
|---|---|---|---|---|
| [37] | 16 | 40 | 20 | feature v conjunction |
| [73] | 23 | up to 54 | 10 | value v no-value |
| [42] | 58 | 40 | 20 | feature v conjunction |
| [74] | 36 | 20 | 10 | feature v conjunction |

Kristjánsson et al [37] was one of the first experiments to investigate multi-target search in humans, with an iPad based paradigm that involved participants 'foraging' for two different types of target (mixed in with two distractor types) by tapping on them sequentially. In the "feature" condition, the targets and distractors were discriminable using just colour as a cue (e.g., the targets were red and green, and the distractors were blue and yellow): in the "conjunction" condition, the categories were defined by both colour and shape (e.g., the targets were red squares and green triangles, and the distractors were red triangles and green squares).

Their experiment used 16 student participants, foraging for 40 targets on each of 20 trials per condition. The dataset used for the following analysis was taken directly from the supplementary material of the journal article: we used trials 6–25 for each participant in each condition, as these were assumed to be the ones used in the original manuscript based on the description in the methods.

Tagu et al [73] is a paper using a foraging paradigm that also includes a manipulation involving value: in the 'value' blocks of the experiment, participants received more points for selecting a high-value target colour compared to low-value target colours. This condition thus manipulated participants' preference for a particular target type (compared to the 'no-value' block where all target types were equal). The task did not require that subjects collect all targets in one trial before moving onto the next trial: instead, they were required to select enough targets to reach a pre-specified number of points. Trials automatically ended once they had reached the required number of points.

We analysed only the mouse foraging blocks, and trials where the targets were randomly distributed (as opposed to clumped). Note that the paper had 3 different target types (one high value and two low value), but we have collapsed these into two target types for convenience. The dataset includes data from 24 participants, with 10 trials for each of the 'value' and 'no-value' blocks per participant, where they could find up to 54 targets on each trial (though they did not always have to find all the targets). For our reanalysis, we used 23 participants (we excluded one participant who did not complete all 10 trials).

Clarke et al [42] contains a near-replicate of [37], except using a larger number of participants: we used 58 participants in our re-analysis (the same number as in the original paper, except that we also removed one additional participant that was labelled as being a different version of the experiment).

Hughes et al [74] is a dataset on a foraging paradigm that includes a scarcity manipulation: in some trials there were unequal ratios of the two target types, whereas in others there were equal ratios (similar to previous experiments such as [42] and [37]). The dataset includes data from 36 participants, each with 10 trials for each of 6 conditions: two levels of difficulty (conjunction and feature) vs. three levels of scarcity (scarce target type A, scarce target type B and equal A & B). As no effects of scarcity were found in [74], we have collapsed across different scarcity conditions in the current analyses.

## 2.6 Analysis plan

An advantage of generative models is that they can be compared and contrasted with human participants and one another over a range of descriptive statistics that represent different aspects of behaviour. Not only does this allow us to

identify areas in which FoMo performs well, it also suggests aspects of human foraging behaviour that are not yet captured by our model.

We start with *accuracy*: for each item selection after the first, we calculate how often our model correctly predicts which item the human participant will select next. Given the nature of the foraging task, baseline (chance) accuracy increases throughout the trial as there are fewer items remaining. We can summarise accuracy across trials to investigate the range of individual differences in predictability, and explore whether some experimental conditions lead to more predictable behaviour.

As well as predicting which specific item will be selected next, we can also summarise our model in terms of its ability to predict when a participant will start a new run of items, as well as other types of commonly used descriptive statistics.

Whereas our accuracy statistics are based on the item-wise (*P*) predictions, the rest of our descriptive statistics are based on the trial-wise *Q* predictions. A summary is given below:

- *Number of runs*: this is computed per trial, using the `rle()` function in R, which calculates the lengths and values of runs of a vector (in this case, the 'item class' label).

- *Maximum run length*: computed as above, but rather than counting the number of runs, we find the value of the longest continuous run in each trial.

- *Percentage Above Optimal (PAO)*: this is computed using the `ForagingOrg` package for R [58]. The optimal path is calculated by estimating the shortest Hamiltonian path that passes through all targets that were collected on the trial. The total length of this path is then compared with the actual path taken by the participant (or model). This is expressed as a percentage, e.g., 0% would be where the actual path corresponds to the optimal path and 100% would be where the actual path was double the optimal.

- *Best-r*: defined as the best absolute Pearson correlation coefficient between the order in which targets are collected and their respective *x* and *y* values [57]. As above, this is computed using the `ForagingOrg` package.

- *intersection rate*: this is computed using the `ForagingOrg` package for R [58]. The metric calculates the number of intersections of non-consecutive trajectories between targets.

- *Lévy flight metric*: This is calculated using the `poweRlaw` package [75]. Briefly, we calculate the exponent $\mu$ that describes the power-law tail of the inter-item distances using a maximum likelihood-based approach [76].

Our aim is to provide a comprehensive account of our model over as many of the commonly used metrics and descriptive statistics as possible. This naturally leads to *a lot* of results and comparisons between different datasets, models, and descriptive statistics. As such, we focus on a selection of the more interesting results in this paper, and more comprehensive accounts can be generated using the code available from https://github.com/Riadsala/FoMo [77]. This study was not preregistered. This GitHub repository also contains full instructions and example workflows for generating simulated data, how to prepare data to be used in FoMo, how to fit different FoMo model variants, and how to use the tools provided to examine model fits.

## 3 Results

In this section we present the results from testing the different versions of FoMo over a range of datasets.

### 3.1 Accuracy for predicting selected items

Overall accuracy is reported in Table 4. We can see that over the course of a trial, all versions of FoMo have an accuracy of around 50% on the majority of datasets while chance performance is below 12% in all cases. That is, the model correctly predicted which items would be selected next around half the time.

**Table 4. Mean test set accuracy for each dataset. The difference in the hance baselines reflects the fact that some experiments involved different numbers of target items. The prior model is based on FoMo v1.0. Note: these overall accuracy scores are relatively insensitive to the differences between models, as accuracy rates vary within trials, between conditions and from person to person, as illustrated in Fig 6.**

| | | FoMo version | | | | |
|---|---|---|---|---|---|---|
| dataset | chance | 1.0 | 1.2 | 1.3 | 1.4 | 1.5 |
| [37] | 0.11 | 0.48 | 0.47 | **0.50** | 0.49 | 0.48 |
| [73] | 0.08 | 0.40 | 0.40 | **0.41** | 0.39 | **0.41** |
| [42] | 0.11 | 0.49 | 0.49 | **0.51** | 0.50 | **0.51** |
| [74] | 0.18 | 0.48 | 0.47 | **0.54** | 0.48 | **0.54** |

These headline figures hide a lot of variation within trials, between experimental conditions, and from one participant to the next. If we compute the accuracy of each participant individually, we find there is considerable variation. For example, in Fig 6 (left, top row), we can see that the model can predict some participants with over 60% accuracy, whereas others are closer to 30% accuracy. These results suggest that while the overall accuracy improvements for Fomo v1.3 (with absolute spatial components included) are relatively modest, using only the averages hides some dramatic improvements for individual participants. In the [74] data, displayed in Fig 6 (left, bottom row), one participant who was one of the worst predicted in FoMo v1.0, with around 30% accuracy, turns out to be highly predictable, over 60%, when analysed with FoMo v1.3. Crucially, the inclusion of the new directional component does not decrease accuracy in any participants.

We can also see variation in predictive accuracy as a trial progresses (see Fig 6, right): as would be expected, accuracy increases as the number of items remaining decreases (though it is worth noting that accuracy is far above chance performance even at the beginning of the trial). It is also interesting to note that v1.3 is consistently more accurate, although the difference between the two models is more pronounced earlier in the trial, suggesting that the directional model is most useful when there are more items available in the display.

Our results suggest that, while the addition of a more complex representation of space only offers marginal benefits to the overall accuracy, for some individuals, they offer a large improvement in our ability to account for foraging behaviour. However, at least in these data, moving from four (v1.3) to eight (v1.4) directions in the mixture model appears to offer decreased accuracy. This likely reflects the fact that most foraging experiments have been conducted on a (slightly jittered) cardinal grid, and therefore four directions are often sufficient to describe behaviour: a strategy that involved diagonal directions would also involve longer distances, and proximity is generally a strong predictor of behaviour in these studies.

Interestingly, relative direction seems to offer only negligible improvements when absolute direction has been added to the model: model version 1.5 includes only the absolute direction parameter (with the relative direction parameter removed) and in three out of four datasets, performs equivalently to model version 1.3 (with relative and absolute direction parameters included) in overall accuracy. The exception to this is [37], which is the only dataset we are using that makes use of 'finger-based' foraging on an iPad (compared to the mouse-based foraging in the other paradigms). While clearly speculative, this may hint at relative vs. absolute directions having different importances based on modality.

### 3.2 Accuracy for predicting switches

We now evaluate our model in terms of predicting switches from a run of one class of item to another. As can be seen in Table 5 FoMo offers good performance - 70% − 80% in most conditions. FoMo compares favourably to the classifier by Le et al [55] on the Kristjánsson et al [37] dataset, especially when considering that this earlier model was trained specifically to detect changes in runs and makes use of a wider selection of features. For conjunction trials our model is within 1.5% accuracy of the classifier, outperforming it when classifying switches and underperforming when detecting repeated selections of the same item. The behaviour diverges more in the feature condition, and the comparison looks quite different

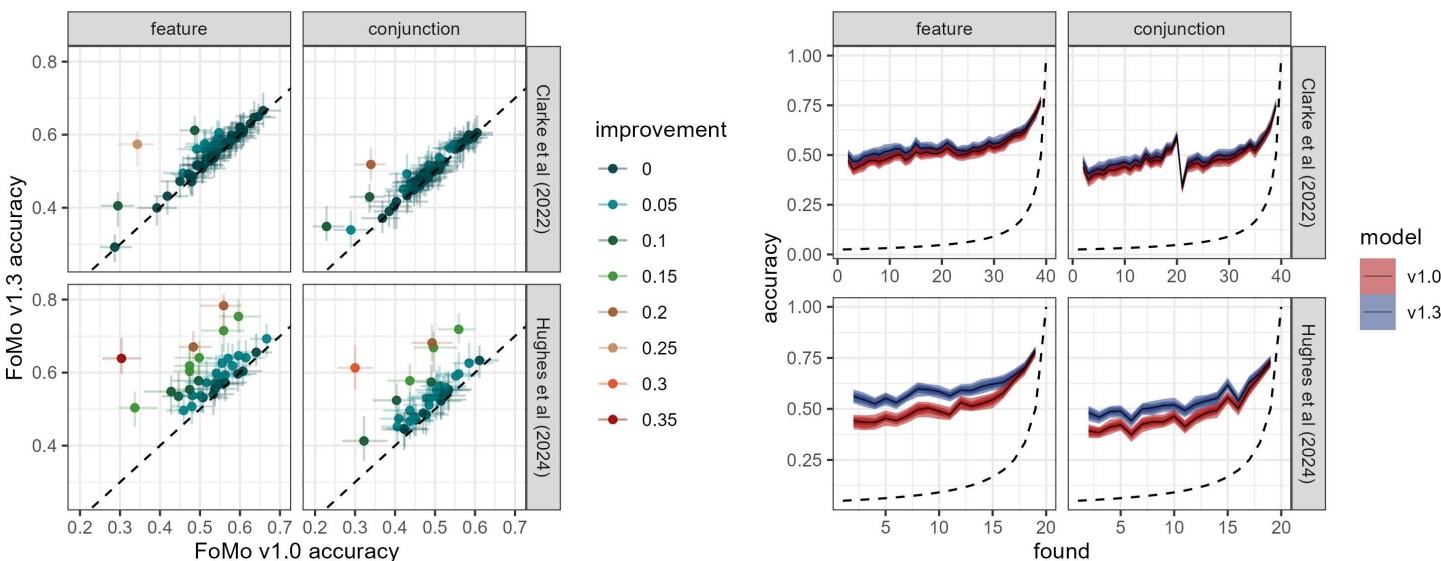

**Fig 6. (*left*): Comparison in accuracy between FoMo v1.0 and 1.3 for the [42] and [74] data.** Each point represents an individual participant, and the crosshairs provide the 95% Highest Posterior Density Interval (HPDI). Colours indicate the improvement in accuracy between models. (*right*): 95% HPDI for accuracy throughout a trial for FoMo v1.3. The dotted line indicates chance performance.

depending on whether we compare to the initial or oversampled model from [55]. Performance for other datasets can be calculated using code available in the Github repository.

### 3.3 Posterior densities

Fig 7 shows the posterior densities for Fomo v1.3 (the model with absolute spatial components included) applied to the Hughes et al [74] dataset (see the Github repository for code to generate plots for other datasets). The parameters for $b_a$, $b_s$, $\rho_\delta$ and $\rho_\psi$ are similar to those reported in v1.0 and show familiar patterns (e.g., the *conjunction* condition has a stronger stick bias but a weaker proximity bias compared to the *feature* condition). The bottom section of Fig 7 shows the strength of the absolute direction parameters in this dataset: generally, horizontal preferences seem to be stronger than vertical ones, and it seems that the biases are generally stronger in the *feature* compared to the *conjunction* condition. This is consistent with behaviour in the *feature* condition being more driven by spatial layout, whereas the *conjunction* trials are more driven by target properties.

### 3.4 Posterior predictions

By using a generative modelling framework, we can also make predictions: i.e., at a given point in any trial, we can ask the model what it thinks that participant will select next. The Bayesian framework of the model also means that we can quantify the uncertainty in this prediction by taking multiple draws from the model posterior. Fig 8 shows some representative examples: the top row shows dramatic improvement for FoMo v1.3 (the model with absolute spatial components included), which is clearly driven by the participant's 'left-right' strategy. The absolute direction component allows the model to account for this strategy, and the model makes quite different (and more accurate) predictions from v1.0 (the baseline model) accordingly. In the bottom row, we see an example where both v1.0 and v1.3 are relatively uncertain about the best option, although in this case model v1.0 is slightly better, perhaps because this participant seems to have a less consistent strategy.

**Table 5. Comparison between FoMo and the classifier developed by [55] on the [37] dataset. We report their intial and final (post feature pruning and oversampling) modelling results, and compare them to FoMo v1.0 and v1.3. The difference coloumn compares performance between v1.3 and the final classifier from [55].**

| | | Le et al (2021) [55] | | FoMo version | | |
|---|---|---|---|---|---|---|
| condition | switch | initial | final | 1.0 | 1.3 | difference |
| feature | stick | 88.0 | 60.7 | 74.4 | 75.2 | 14.5 |
| feature | switch | 51.3 | 88.1 | 69.3 | 70.5 | -17.6 |
| conjunction | stick | 99.4 | 93.5 | 92.0 | 92.2 | -1.3 |
| conjunction | switch | 46.7 | 75.7 | 76.4 | 76.8 | 1.1 |

As discussed above, an advantage of using a generative modelling framework is that we can assess the extent to which FoMo can account for foraging behaviour over a range of metrics commonly used in the literature. In Fig 9, we give an overview of how well FoMo v1.3 is able to predict these metrics across the range of datasets. In general, we can see that FoMo is doing a good job, with its outputs correlating well with all metrics (and in some cases lying almost directly on the identity line). Some of the poorer fits include the maximum run length for the Tagu et al dataset [73]: this is likely because this is an inexhaustive search paradigm, and the model currently assumes exhaustive foraging. The predictions are also often slightly biased for best-*r* and PAO metrics. While the predicted and observed values generally correlate well, the model systematically predicts lower best-*r* values and higher PAO values than are seen in the data, suggesting that humans remain more 'optimal' than the model, perhaps through the use of more systematic strategies. We can also see that model v1.3 does a better job of predicting these metrics on aggregate than v1.0 (Fig 11, top: note we have not included the Tagu et al dataset [73] as the model will predict beyond the observed data, confounding error metrics).

### 3.5 Fixing the first item selection

One factor that affects trialwise predictions is the initial starting point. In particular, we know that the real starting positions in visual foraging datasets are not random, with most participants beginning in the top left-hand corner or in the centre of the screen [42,78]. However, these biases are not currently implemented in FoMo, and clearly, if the model starts at a different item to human participants, the overall path is likely to be quite different. We are now able to run model versions which 'fix' the first item selection (i.e., tell the model which target the participant selected first) to assess how much this improves model performance. Fig 10 shows examples of this. When the starting position is fixed (top row), the model gets the first 8 selections completely correct in one prediction, and generally the top row of targets are selected early in the trial (similar to the participant's real behaviour). The increasing variability in the start position shown in the bottom row leads to more variability in the selections, meaning that while the overall patterns still look similar, trialwise accuracy is likely lower. Across all predictions, Fig 11 (bottom) suggests that the fixed start point also improves predictions of the foraging metrics, as if we take the scaled difference between observed and predicted measures (so that they are comparable) and then average, we can see that the fixed start point model has lower average error for most metrics.

### 3.6 Different $\kappa$ values

$\kappa$ is passed as a hyperparameter to our models with absolute directions included: for the basic version of model v1.3 (the model with absolute spatial components included), its value was set as 20. However, to check that this is an appropriate value for $\kappa$, we re-ran the models for v1.3 on the Hughes et al dataset [74] with a range of other $\kappa$ values ($\kappa$ = 10, 25 and 50). There were no strong differences in overall accuracy for these different values, and thus we used $\kappa$ = 20 for the other datasets. However, it is not entirely clear to us how the best $\kappa$ value might differ across different datasets, particularly

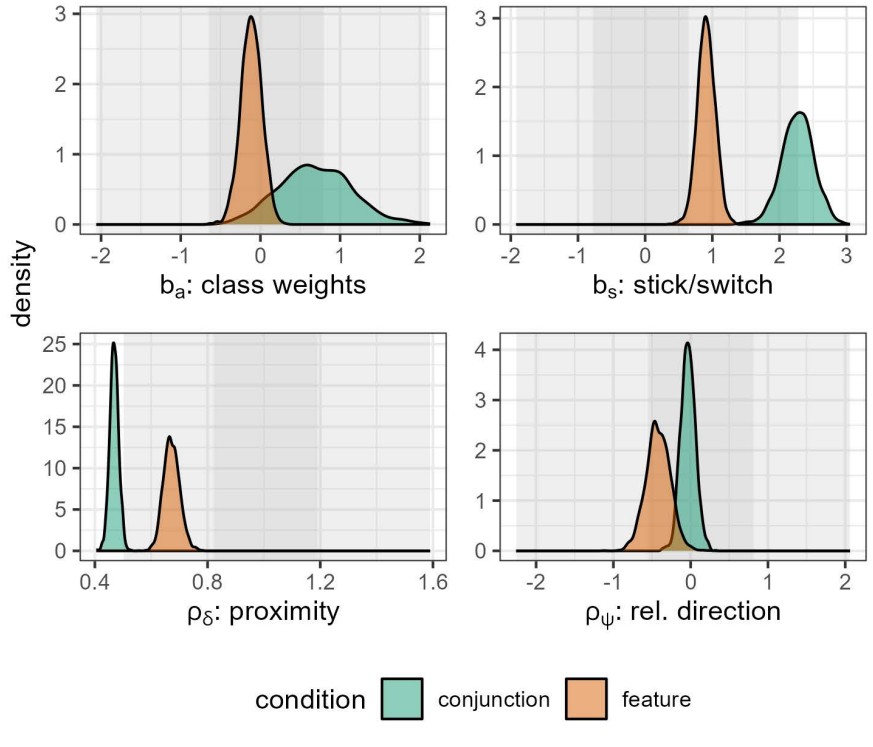

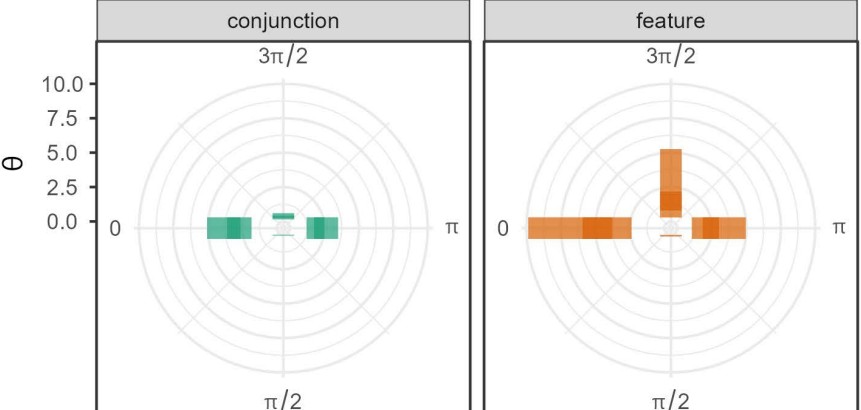

**Fig 7. Posterior densities for FoMo v1.3, applied to the [74] dataset.** (*top*): Posterior estimates for the four core FoMo parameters. (*bottom*): Illustration of the 4 $\theta_k$ direction weights, for each of our two conditions. The shaded bar indicates the 97%HPDI, with the dark inner region giving the 53% interval. We can see that in both conitions there is a preference for horizontal directions, and stronger directional effects in the feature condition.

when moving away from the types of 'screen-based' paradigms we have used in this manuscript (that likely encourage cardinal grid-following behaviour) and therefore we encourage other users of FoMo to carry out their own sensitivity analyses.

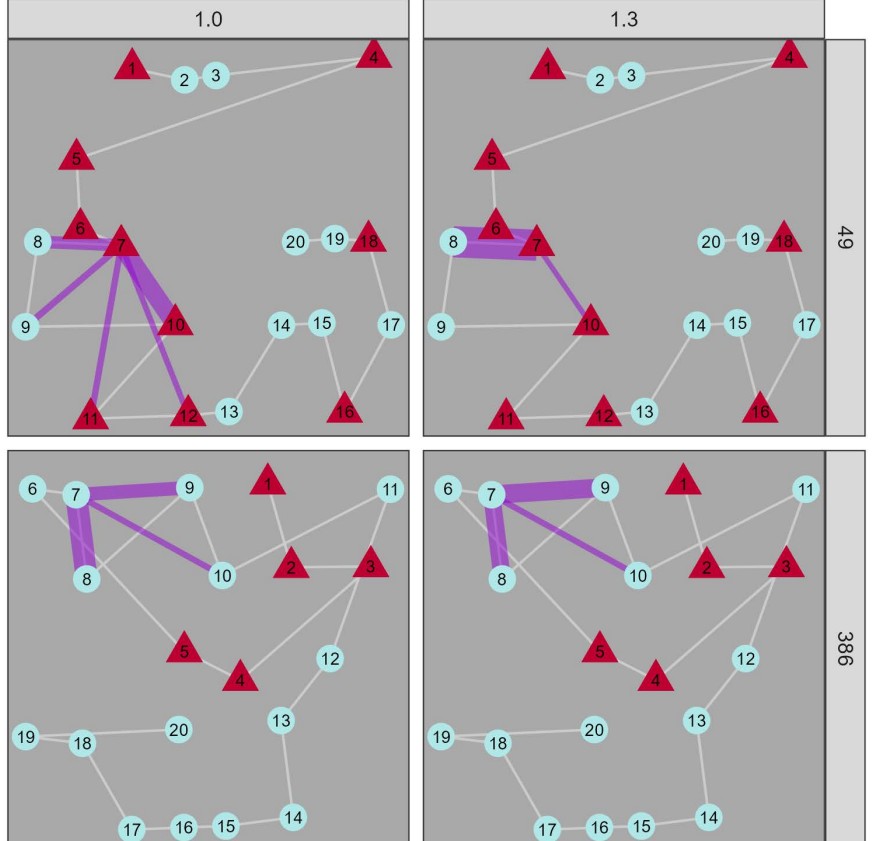

**Fig 8. Comparison of simulated paths from FoMo v1.0 and v1.3 with observed data from a human participant from the [74] dataset.** The items are illustrated as red triangle and pale blue circles, and the white line illustrates the order in which the human particpant selected the items. The purple lines indicate the weightings that the model assigns to each possible next target (thicker means more weight).

## 4 Discussion

FoMo aims to predict visual foraging behaviour on a target-by-target basis. In the current manuscript, we demonstrate that FoMo is remarkably successful in this aim, achieving an average accuracy of 40–50% for all the datasets we test (compared to a much lower average chance level of around 10–20%, depending on the exact dataset). Importantly, incorporating a parameter for absolute direction into the model improves predictions compared to the 'baseline' model with only relative direction included, and we show this is largely because a small subset of participants show a strong preference for absolute directions, and the model is therefore able to get a lot better at predicting these individuals. The modularity of our modelling approach means that we can also start to ask questions about the most parsimonious set of parameters to predict behaviour: for example, we show that for many (but not all) of our datasets, including absolute direction means that the relative direction parameter is redundant, at least in terms of improving average accuracy. Finally, we believe FoMo can act as a unifying framework for visual foraging research, as it is able to predict a wide range of descriptive statistics that have been used in previous work.

### 4.1 Outliers and individual differences

One strength of our approach is for characterising individual differences: our model can deal with people who show 'outlier' behaviour, and can actually tell us something about why those people are outliers. For example, in the Hughes et al

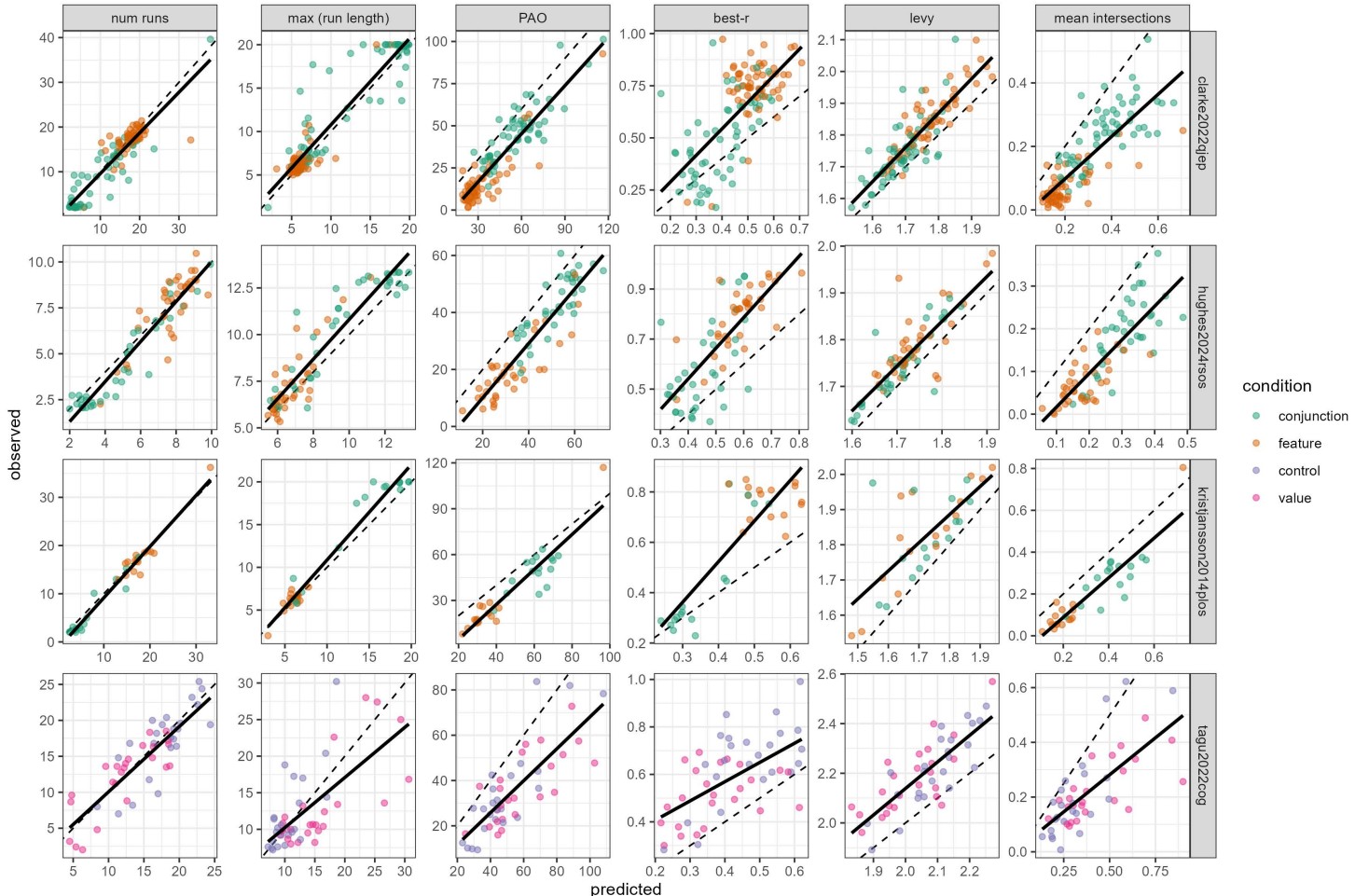

**Fig 9. Comparison of predicted and observed path statistics (FoMo v1.3) over a range of datasets.** Each dot represents an individual participant in a condition, the dotted line indicates the $y = x$ identity line, while the solid black lines give the best fit linear regression. Note that the poorer performance for the [73] is almost certainly due (at least in part) to our use of an exhaustive foraging model to account for an inexhastive paradigm.

dataset [74], we found a number of participants for whom including an absolute direction parameter starkly improved the model's prediction of their behaviour. We think this is particularly exciting because it does not seem to matter that this is relatively rare behaviour: the model predictions improve, without affecting predictions for participants for whom absolute direction matters less. Often modelling approaches deal poorly with outliers or 'rare' events, but FoMo seems to be able to incorporate them well. This modelling approach also provides a principled way to tackle the idea of 'strategies', which are often quite hard to define: for example, we could determine that participants whose model prediction accuracy increases more than a certain percentage when tested with model v1.3 (compared to model v1.0) are using an 'absolute direction' strategy. This has the potential to be very powerful for developmental and clinical work, where we may want to be able to group participants based on characteristic behaviours.

It is interesting to note that we see bigger improvements as we move from model version 1.0 to 1.3 for some datasets compared to others: for example, the Clarke et al dataset [42] shows a smaller increase than the Hughes et al dataset [74]. Hughes et al [74] involved a 'scarcity' based task where on some trials, one target was less common than the other. We had originally hypothesised that participants might show a scarcity bias, preferring to collect the rarer targets first,

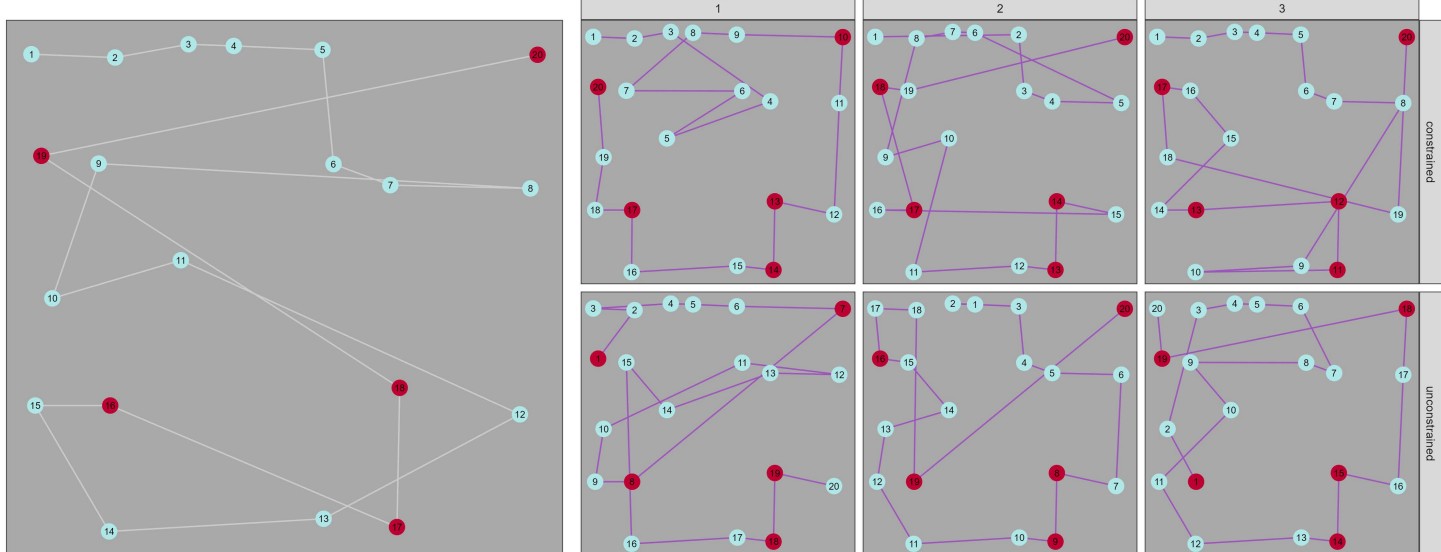

**Fig 10. The items are illustrated as red triangle and pale blue circles.** (*left*) The while line illustrates the order in which the human particpant selected the items in a trial from the Hughes et al dataset [74]. (*right*) Six example predictions, shown in purple, of predictions from FoMo v1.3. The three plots in the top row represent simulations in which we have constrained the initial item selection to match the item selected by the human participant on this trial. The three plots on the bottom are simulations in which the initial item selection is unconstrained.

which we did not find evidence for in our original manuscript, but it does seem that at least for a subset of participants, this manipulation may have pushed them into a scanning-based strategy to ensure they had found all the 'rare' targets. In addition, it is likely that the structure of the foraging stimulus and the instructions about strategy that are given to participants will strongly affect behavioural patterns: for example, to date, most visual foraging studies have used 'grid-like' 2D stimuli that are likely to encourage participants to use an absolute-direction-based strategy, which may be less evident if targets are found in patches, or if participants must forage in a real-world environment where not all targets are immediately visible. Eye and head movements are also likely to become more important in paradigms that move away from screen-based foraging. We therefore think it may be unrealistic to expect the parameter values in our model to look similar for each foraging dataset, but with more data, we may be able to understand better which environmental conditions change the importance of each parameter, and which parameters are most likely to see large individual differences.

The foraging literature to date has used a wide range of different metrics for evaluating performance, from run-length statistics (e.g., [37]) to scanpath metrics such as best-*r* (e.g., [58,79]) or Lévy flight metrics (e.g., [76]). We show here that by using predictions from the FoMo model, we can generate very good approximations to the true empirical values for essentially all of these metrics. We would therefore argue that FoMo can act as a unifying theory: the model is not explic-itly designed to account for these metrics, and yet it is able to make good estimations in most cases, suggesting that it is capturing something fundamental about how behaviour unfolds in this task.

## 4.2 Limitations and future directions

Although our model is capable of making good predictions of human behavior in a foraging task, there is clearly still room for improvement. One challenge is that some strategies on this task limit the observation space: for example, if a partic-ipant always works top-to-bottom to find targets, the model never gets a chance to 'observe' their behaviour when there are potential target items above their current ($x$, $y$) position in the display (as these have already all been selected). A consequence of this is that the model's corresponding $\theta$ parameter is now unconstrained, limiting our ability to make good

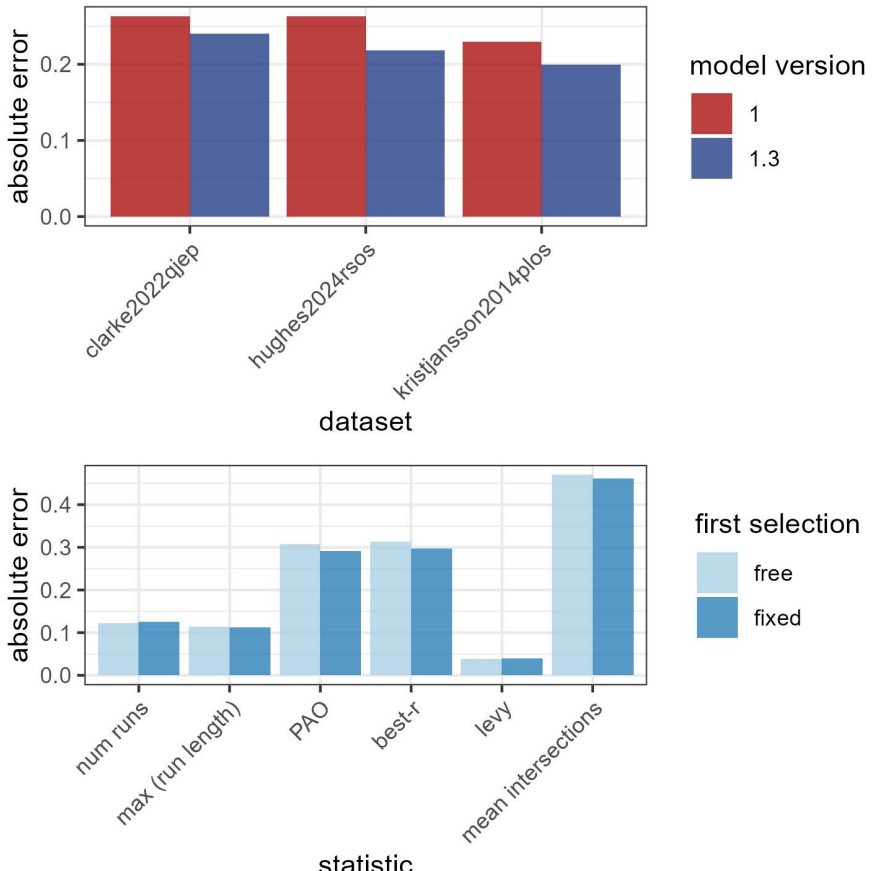

**Fig 11. (*top*): estimate of scaled absolute error averaged across first selection variants and statistics, for each dataset and model version.** (*bottom*): estimate of scaled absolute error averaged across datasets for model v1.3, for fixed and free first selection and for each statistic.

predictions in novel situations. Similarly, one common strategy is a 'reading-type' strategy where participants scan targets left-to-right and then do a 'carriage return' back to the left side of the screen to begin again, which the model cannot easily predict. While it would be possible to include ways to account for this strategy, it is more difficult to envisage a method that would also deal well with non-screen-based tasks, where we predict this behaviour would be less prevalent.

We used von Mises mixture models to implement absolute direction parameters into FoMo, which involves adding $\theta$ and $\kappa$ parameters into our model. However, it can be challenging to fit these parameters simultaneously: in particular, you can end up with one parameter compensating for the other, essentially leading to $\kappa$ being unconstrained (as if $\theta$ is small, $\kappa$ can be large, and vice versa). We circumvent this issue by using a hyperparameter for $\kappa$. Our testing suggests that the exact value of $\kappa$ has a relatively small effect and does not influence any of the conclusions we draw. However, it is possible that a more sophisticated modelling approach with more tailored priors could avoid the need to use hyperparameters in this way.

FoMo is currently designed to predict target-by-target selection in exhaustive foraging paradigms. It does a reasonable job at predicting inexhaustive search in the Tagu et al dataset [61], but the accuracy is notably lower compared to the other datasets considered in this manuscript; though since we have only one inexhaustive dataset, it is unclear whether this lower accuracy is due to this, or other unique aspects of that dataset (such as its value vs. no value manipulation). However, broadening FoMo to be able to work better with inexhaustive foraging by including a parameter for when to 'give up' on a given area of targets would also allow the model to make predictions for 'patch leaving' paradigms, helping to

bridge the gap between these two types of foraging experiment that have normally been studied separately. This would be particularly exciting given recent evidence that spatial factors may predict patch leaving behaviour better than classic intake rates used in MVT [79].

Finally, our work on FoMo to date largely considers two-dimensional, static target collection environments (although see [80] for an example of implementing FoMo in an experiment that used first-person avatars in a virtual environment). A future goal for FoMo is to understand how well it can predict behaviour in more realistic environments, including VR-based tasks [81] and tasks where targets move dynamically [13]. We believe that the basic principles underlying the model are likely to remain constant, although the importance of particular parameters may vary; for example, we might expect relative direction to be more important where participants have to physically walk through an environment, as momentum to keep going in the same direction is likely to be strong. Real-world environments may also require new parameters, such as heading direction, to help explain behaviour, but as demonstrated here, these are easy to implement and test in FoMo.

## 4.3 Conclusion

We have updated our model of foraging, FoMo, to incorporate absolute direction parameters and have shown that we can make good predictions of target-by-target over a range of datasets. Futhermore we have demonstrated that our model can account for human behaviour over a wide range of secondary metrics and summary statistics that have previously been used in the foraging literature. We would therefore argue that FoMo is the first visual foraging model that can be considered 'unifying', in that it can bring together data and approaches from different research groups and explain them using a single framework. This approach is exciting both for studying foraging and for wider cognitive psychology research for several reasons. Firstly, we believe that emphasising prediction in psychological theories is important as it provides a good way of truly measuring how successfully we are answering the question we actually want to know about, i.e., how well can we predict behaviour? Furthermore, we have designed the model to be modular in order to facilitate hypothesis testing, i.e., does a given parameter affect behaviour in my paradigm? Finally, we have aimed to make the model as accessible and open source as possible to encourage other researchers to attempt to use it - and break it! - with their own data. By finding situations in which FoMo fails to make good predictions, we can learn more about which aspects of foraging behaviour are still poorly explained and gain insight about how to improve our modelling efforts. Overall, we aim to demonstrate how a formal model can be iteratively applied, evaluated, and improved, and we believe that our approach may have broader applicability for model development in other areas of psychology.

## 4.4 CRediT Author Contributions

| Role | Contributor |
| --- | --- |
| Conceptualization | AC & AH |
| Data Curation | AC & AH |
| Formal Analysis | AC & AH |
| Funding Acquisition | *NA* |
| Investigation | AC & AH |
| Methodology | AC & AH |
| Project Administration | AC & AH |
| Resources | AC & AH |
| Software | AC & AH |
| Supervision | AC & AH |
| Validation | AC & AH |
| Visualization | AC & AH |
| Writing – Original Draft Preparation | AC & AH |
| Writing – Review & Editing | AC & AH |

## Author contributions

**Conceptualization:** Alasdair D. F. Clarke, Anna E. Hughes.

**Data curation:** Alasdair D. F. Clarke, Anna E. Hughes.

**Formal analysis:** Alasdair D. F. Clarke, Anna E. Hughes.

**Investigation:** Alasdair D. F. Clarke, Anna E. Hughes.

**Methodology:** Alasdair D. F. Clarke, Anna E. Hughes.

**Project administration:** Alasdair D. F. Clarke, Anna E. Hughes.

**Resources:** Alasdair D. F. Clarke, Anna E. Hughes.

**Software:** Alasdair D. F. Clarke, Anna E. Hughes.

**Supervision:** Alasdair D. F. Clarke, Anna E. Hughes.

**Validation:** Alasdair D. F. Clarke, Anna E. Hughes.

**Visualization:** Alasdair D. F. Clarke, Anna E. Hughes.

**Writing – original draft:** Alasdair D. F. Clarke, Anna E. Hughes.

**Writing – review & editing:** Alasdair D. F. Clarke, Anna E. Hughes.

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
