## [Decision Letter · Decision Letter 0]

17 Feb 2026

PCOMPBIOL-D-25-02573

FoMo: A Unifying Theory of Visual Foraging

PLOS Computational Biology

Dear Dr. Clarke,

Thank you for submitting your manuscript to PLOS Computational Biology. After careful consideration, we feel that it has merit but does not fully meet PLOS Computational Biology's publication criteria as it currently stands. Therefore, we invite you to submit a revised version of the manuscript that addresses the points raised during the review process.

We look forward to receiving your revised manuscript.

Kind regards,

Alejandro Tabas, Ph.D.

Academic Editor

PLOS Computational Biology

Natalia Komarova

Section Editor

PLOS Computational Biology

**Journal Requirements:**

At this stage, the following Authors/Authors require contributions: Alasdair D. F. Clarke, and Anna E Hughes. Please ensure that the full contributions of each author are acknowledged in the "Add/Edit/Remove Authors" section of our submission form.

4) Your manuscript is missing the following sections: Methods.  Please ensure all required sections are present and in the correct order. Make sure section heading levels are clearly indicated in the manuscript text, and limit sub-sections to 3 heading levels. An outline of the required sections can be consulted in our submission guidelines here:

5) Please upload all main figures as separate Figure files in .tif or .eps format. For more information about how to convert and format your figure files please see our guidelines:

**Reviewers' comments:**

Reviewer's Responses to Questions

**Comments to the Authors:**

Reviewer #1: Generally speaking I have been enthusiastic about the approach that this group uses and the modelling approach that they presented recently. They now provide a fine addition to their previous modelling. This is for sure a nice continuation of their work until now, that continues their previous work in a logical way.

So at this point, I do not have many comments, and I think that this is a good contribution to the literature. I also think that their coverage of the literature is quite comprehensive, but I have comments, however (see below)

Here, the authors add a spatial component to their previous model and that is why the authors call this a “unifying” model of foraging.

The idea behind the model being “unifying” is that it can explain data from differing paradigms. There is a very important component missing, if this is to be comprehensive, however. Any forager in the real world will face a constantly dynamic, changing scene. Of course, this adds massive computational complexity, but this, nevertheless cannot go unaddressed (see below).

I would encourage the authors to speculate, if not implement how their model would work in a real-world setting. They do rightfully mention that foraging tasks are a pretty good analogue to real-world scenarios. For humans, the main tasks involve gathering information, the frontal lobe equivalent of more basic tectum-based chicken foraging.

While I do not claim that the authors are stuck within their own paradigm, I do encourage them to try to take this to more general scenarios. Let’s say I try to cross a busy street. For this, I need to gather lots of information (forage for information), how does the model help us understand how that might occur? The “problem” is that these are non-dynamic scenes. Foraging during continuously changing scenes, must therefore be on the menu? I highlight Thornton et al. (https://link.springer.com/article/10.1186/s41235-021-00299-w; see also https://journals.plos.org/plosone/article?id=10.1371/journal.pone.0219827) as an important consideration – these paradigms get closer to real world scenarios, in my opinion, and that is what the authors in the current manuscript are after. I am not asking the authors to explicitly model theses sort of paradigms, but to speculate what the next steps could look like.

One minor point but could be important is that the authors refer to summary statistics regarding previous treatment of foraging results. The term “summary statistics” is quite loaded in the literature on visual cognition. So I simply want to warn the authors that many people may think of how ensembles are summarized in perception when they hear this term.

Reviewer #2: All attached

Reviewer #3: This is an interesting paper describing the next version of the authors’ FoMo (Foraging Model). Version 1.3 includes a directional component that helps in the prediction of what item gets picked next in a foraging task. The model does a very decent job of predicting behavior in several datasets. Someone other than me should vouch for the computational details of the model but it looks solid to me.

I have a few issues, large and small

1) My main concern is that FoMo is, at present, a big hammer that is hitting a rather small nail. There is a lot of modeling here for purposes of explaining a rather restricted set of data. FoMo “currently assumes exhaustive foraging”. This is fine for analyzing Kristjansson-style data where Os are instructed to search exhaustively for two types of target. I suspect that non-exhaustive foraging is much less systematic and, indeed, apparently FoMo does not do as well with the Tagu data which (I gather) is non-exhaustive. So, I worry if FoMo might be a lot of model for showing that Os tend to search very systematically when forced to find everything.

2) At the very least, you should be saying more about the distinction between exhaustive and non-exhaustive search tasks. For instance, in section 2.5, you should be more explicit about the instructions in the tasks that are being modeled I don’t think you mention that the Tagu task is non-exhaustive. That point comes up later.

3) Switching behavior is also quite different (I think) in non-exhaustive foraging: e.g. Wolfe, J. M., Cain, M. S., & Aizenman, A. M. (2019). Guidance and selection history in hybrid foraging visual search [journal article]. Atten Percept Psychophys, 81(3), 637-653. https://doi.org/10.3758/s13414-018-01649-5

4) Oh, and that paper looks at switching between four target types. How would FOMO do with that?

5) I wonder how well a really simple nearest neighbor model do with standard Kristjansson data? Such a model would have a switch probability (low for conj, higher for feature) and then just a simple nearest neighbor rule. That might be a nice benchmark to show how FoMo can do.

Some smaller points

6) P3 says , about our driller/scanner work; “radiologists who adopted a ’scanning’ strategy, searching an entire 2D scan before moving in depth performed more poorly, making more search errors, compared to participants who had a ’drilling’ strategy, restricting their eye movements to one portion of the image and scrolling through in depth (37).” I did not go back to check, but my recollection is that you are right in numerical terms but that the difference is not statistically reliable. Hard to get the required power with radiologist observers.

7) On P7, I could have used a few more words justifying the negative exponential

8) And on P8 “Constraining K”, I could use some more explanation about the added parameters. What are they doing?

9) For the FoMo description, I would add a table of terms as an appendix or something. I can’t keep them in memory. (b(a), b(s)….etc)

10) Fig 5 is very small and hard to read.

11) P14 says “with most participants beginning in the top right-hand corner” Not top left as in reading? See figs 1 and 7?

In sum, the FoMo project is a good and rigorous one. I think the current version is a bit limited in its scope. I hope that FoMo2 predicts quitting times, for example. Nevertheless, FoMo represents one of the best efforts to formally model human foraging behavior.

Thanks

Jeremy Wolfe (obviously signed review)

**Have the authors made all data and (if applicable) computational code underlying the findings in their manuscript fully available?**

Reviewer #1: None

Reviewer #2: Yes

Reviewer #3: Yes

PLOS authors have the option to publish the peer review history of their article (what does this mean?). If published, this will include your full peer review and any attached files.

Reviewer #1: No

Reviewer #2: **Yes:** Beatriz Gil-Gómez de Liaño

Reviewer #3: No

**Figure resubmission:**
---

## [Decision Letter · Decision Letter 1]

16 Apr 2026

PCOMPBIOL-D-25-02573R1

FoMo: A Unifying Theory of Visual Foraging

PLOS Computational Biology

Dear Dr. Clarke,

Thank you for submitting your manuscript to PLOS Computational Biology. After careful consideration, we feel that it has merit but does not fully meet PLOS Computational Biology's publication criteria as it currently stands. Therefore, we invite you to submit a revised version of the manuscript that addresses the points raised during the review process.

We look forward to receiving your revised manuscript.

Kind regards,

Alejandro Tabas, Ph.D.

Academic Editor

PLOS Computational Biology

Natalia Komarova

Section Editor

PLOS Computational Biology

**Journal Requirements:**

1) Please ensure that the affiliation of the author Clarke is listed on in the "Affiliation" tab in the online submission form.

2) Please ensure that the figures are uploaded in a correct numerical order in the online submission form.

**Reviewers' comments:**

Reviewer's Responses to Questions

Reviewer #1: I have no further comments and recommend publication

Reviewer #2: Thanks for all changes, I think the paper is ready to be published.

Reviewer #3: This is a fairly lightly revised paper describing the latest version of the detailed FOMO model of human visual foraging. I still wish that the model had been deployed on a richer set of phenomena. The discussion in the Response to Review is quite interesting, pointing in more wide-ranging directions than are found in the revisions to the paper. Nevertheless, the existing paper will be a useful documentation of the current state of an evolving model.

A few minor bits

P6 says These four features are illustrated in Figure 3. These are more formally defined in Section 2.1, ….” Are those the right section #s? I think you mean maybe 3.2.

P 11 says “This dataset is inexhaustive; participants had to collect a certain number of points on each trial, but this did not necessarily correspond to collecting all targets on the screen.”

This makes it sound like the dataset is incomplete. Maybe something like “The Tagu et al task did not require that Os collect all targets in one patch before moving on to the next patch. They were required to collect (at least??) N points per patch”

Now that I can see Fig 6, I don’t fully understand it. The numbers on the axes on the lefthand figure are accuracy? The secondary Y-axis is a reference to a dataset? And what is HPDI? (not High-Pressure Direct Injection, I am guessing)

Jeremy Wolfe

**Have the authors made all data and (if applicable) computational code underlying the findings in their manuscript fully available?**

Reviewer #1: None

Reviewer #2: None

Reviewer #3: None

PLOS authors have the option to publish the peer review history of their article (what does this mean?). If published, this will include your full peer review and any attached files.

Reviewer #1: No

Reviewer #2: **Yes:** Beatriz Gil-Gómez de Liaño, Ph.D.

Reviewer #3: **Yes:** Jeremy Wolfe (But I think publishing reviews is not a good idea)

**Figure resubmission:**
---

## [Editor Report · Decision Letter 2]

23 Apr 2026

Dear Clarke,

We are pleased to inform you that your manuscript 'FoMo: A Unifying Theory of Visual Foraging' has been provisionally accepted for publication in PLOS Computational Biology.

Best regards,

Alejandro Tabas, Ph.D.

Academic Editor

PLOS Computational Biology

Natalia Komarova

Section Editor

PLOS Computational Biology

---

## [Editor Report · Acceptance letter]

PCOMPBIOL-D-25-02573R2

FoMo: A Unifying Theory of Visual Foraging

Dear Dr Clarke,

I am pleased to inform you that your manuscript has been formally accepted for publication in PLOS Computational Biology. Your manuscript is now with our production department and you will be notified of the publication date in due course.

With kind regards,

Lilla Horvath
